# Factors Influencing the Levels of Awareness on Malaysian Healthy Plate Concept among Rural Adults in Malaysia

**DOI:** 10.3390/ijerph19106257

**Published:** 2022-05-21

**Authors:** Norsyamlina Che Abdul Rahim, Mohamad Hasnan Ahmad, Cheong Siew Man, Ahmad Ali Zainuddin, Wan Shakira Rodzlan Hasani, Shubash Shander Ganapathy, Noor Ani Ahmad

**Affiliations:** Institute for Public Health, National Institutes of Health, Ministry of Health Malaysia, Setia Alam 40170, Malaysia; hasnan.ahmad@moh.gov.my (M.H.A.); smcheong@moh.gov.my (C.S.M.); ahmadali@moh.gov.my (A.A.Z.); wshakira@moh.gov.my (W.S.R.H.); dr.shubash@moh.gov.my (S.S.G.); drnoorani@moh.gov.my (N.A.A.)

**Keywords:** level of awareness, Malaysian Healthy Plate, rural, adult

## Abstract

Malaysian Healthy Plate was launched in 2017 as a vital component of the Malaysian Ministry of Health’s “Quarter-Quarter-Half” program. It is predicted that this concept will bring positive change to the citizen’s dietary habits; however, the residents in rural areas may experience less exposure to this campaign, or lack of understanding to the concept of Malaysian Healthy Plate. Hence, this study aims to assess factors affecting the campaign’s awareness among the rural population in Malaysia. The National Health and Morbidity Survey (NHMS) 2019 focused on Non-Communicable Diseases (NCDs). Data collection was carried out from July to September 2019. Questionnaires that pertained to awareness, knowledge, and practice were included in this nationwide survey. Data collected from adults in rural areas aged 18 years old and above were used as respondents for the statistical analysis. Complex sample multiple logistic regression analysis was conducted to determine the association between the independent variables and awareness of the Malaysian Healthy Plate concept. About four fifths (83.2%) of rural adults in Malaysia were unaware of the Malaysian Healthy Plate concept after three years of implementation. The unawareness was significantly higher in males (91.3%), adults aged above 60 years old (91.8%), adults of others ethnicity (88.5%), those without formal education (95.2%), widows/widowers/divorcees (88.1%), retirees/adults who were not working (88.4%), and household income at the bottom 40% (B40) (85.0%). Unawareness of this concept was significantly associated with male gender (aOR = 4.12; 95% CI: 3.06–5.56); age, 40–59 years (aOR = 1.46; 95% CI:1.08–1.97); without formal education (aOR = 3.47; 95% CI:1.34–9.01); working in private sector (aOR = 2.75; 95% CI: 1.59–4.77); self-employed (aOR = 2.78; 95% CI: 1.58–4.87); retirees (aOR = 2.32; 95% CI: 1.23–4.36); and unpaid workers (aOR = 2.61; 95% CI: 1.51–4.51). Awareness of the Malaysian Healthy Plate concept is associated with rural adults being males, with older age, with lower socio-economic status, without partner, and without job. This study suggests that a more effective strategy is needed to increase the awareness of the Malaysian Healthy Plate concept among rural adults.

## 1. Introduction

The World Health Organization (2014) [1] estimated that 73% of Malaysian deaths were due to Non-Communicable Diseases (NCDs). The National Health and Morbidity Surveys [2,3,4] showed an increasing trend of prevalence of diabetes mellitus from 11.6% in 2006, to 13.4% in 2015, to 18.3% in 2019. In contrast, the prevalence of hypercholesterolemia increased from 20.7% in 2006 to 47.7% in 2015. The most important risk factors for NCDs include high blood pressure, high blood cholesterol concentrations, inadequate intake of fruits and vegetables, overweight or obesity, physical inactivity, and tobacco use [5]. In addition, overweight and obesity, a major public health problem, is a major risk factor for universal death. Globally, there has been an increase in the incidence of obesity among adults [6,7]. In 2014, the WHO [1] estimated that more than 1.9 billion adults aged 18 years old and above were overweight, and over 600 million adults were obese. In Malaysia, a total of 50.1% of adults were overweight or obese [4]. The World Health Organization showed that Malaysia has the highest prevalence of overweight people (BMI > 25 kg/m^2^) compared to other countries in Southeast Asia [8]. Several risk factors, such as family history, diet, sedentary lifestyle, smoking habits, and other environmental factors, increase the risk of NCDs. Most NCDs are considered preventable because they are caused by modifiable risk factors, such as unhealthy diet and sedentary lifestyle.

In order to relieve the increasing burden of NCDs and obesity in Malaysia, the Ministry of Health Malaysia introduced Malaysian Dietary Guidelines (MDG) in 1999 [9]. MDG is used as a reference or guide to encourage Malaysians to adapt a balanced diet that consists of various types of food in moderate amounts. Along with the formation of the MDG, the Malaysian Food Pyramid was also introduced in 1999. The Malaysian Food Pyramid is formed as a guide for choosing the type and amount of food for the public in promoting a balanced, moderate, and varied diet [10]. Malaysian Healthy Plate was produced in 2017 as a complementary tool for the Malaysian Food Pyramid. It shows the balanced single meal by visualizing the recommended amount of food to be eaten on a plate (Figure 1) [11]. The Malaysian Healthy Plate consumption of a quarter plate of grains, grains products, cereals products, and tubers; a quarter plate of fish, poultry, meat, and legumes; and half a plate of fruits and vegetables. In addition, plain water is suggested to replace the intake of sugary drinks [12].

The Malaysian Healthy Plate was profoundly promoted to the public in different settings, such as health clinics, schools, and grocery markets, through mass media and an online application, with a message of *Suku-Suku-Separuh* (translated as “Quarter-Quarter-Half”). The healthy plate concept is more tangible and relevant than the more abstract food pyramid, making it easier to understand and implement. It is predicted that this concept will positively change citizens’ dietary habits. However, the Malaysian may experience less exposure to this campaign, or lack understanding of the Malaysian Healthy Plate concept. Continuous education and promotion through various channels, and multiagency collaboration on the implementation, are recommended to ensure this healthy eating concept can be delivered effectively to the public, especially now during the COVID-19 pandemic.

The awareness of the Healthy Plate concept was found to be 20.4% in the population, with only 14.0% of the adults who were aware of it admitting to practicing it daily [4]. This indicates that four out of five Malaysian adults were unaware of the concept of Malaysian Healthy Plate. The report also showed that respondents from the urban areas (21.5% (95% CI: 19.64, 23.43)) were more aware of the Malaysian Healthy Plate concept compared to the rural areas (16.8% (95% CI: 14.74, 18.98)) [4]. In achieving the Sustainable Development Goals (SDGs) that lay the foundations of leaving no one behind, the study chose the rural population to inculcate the planning that aligns with the global drives to reduce socio-economic inequalities in health. The communities living in rural locations give very minimal importance and priority to the routine medical screening because of their perceived good self-health [13]. An increase in NCDs’ risk factors, including smoking, inactive life (sedentary or less physical activity), obesity, and poor dietary habits, was observed in Malaysians from rural areas [14].

To evaluate the effectiveness of the Malaysian Healthy Plate campaign in rural areas, a thorough assessment of people’s awareness, knowledge, and practice of the Malaysian Healthy Plate concept is needed. In addition, this study also aims to identify the factors influencing the awareness of the Malaysian Healthy Plate concept among rural adults in Malaysia. The findings of this study are important to provide information of the effectiveness for the Ministry of Health Malaysia to produce evidence-based policies, and review the implementation status. It is presumed that it will assist the Ministry of Health Malaysia in preparing and executing a successful community health promotion program.

## 2. Materials and Methods

### 2.1. Study Design and Sampling Procedure

This study used data from the National Health and Morbidity Survey (NHMS) 2019, a nationwide community-based cross-sectional study. NHMS 2019 focused on NCDs and Health Care Demand (HDC). All individuals aged 18 years old and above, and who stayed for at least two weeks in the selected living quarters, were invited to participate in NHMS 2019. Individuals aged below 18 years old, or adults who were severely ill at data collection, were excluded from this study. The sample size was calculated using the single proportion formula with adjustment by considering the complex design, number of strata, and expected non-response rate. Based on a proportion of 50%, as this is the first time an assessment on the Malaysian Healthy Plate concept has been tested, type 1 error or alpha (α) equal to 0.05, power (β) equal to 80%, design effect equal to 1.5, and the number of strata equal to 16, a total of 9216 individuals above 18 years old were required for analysis.

The National Health and Morbidity Survey (NHMS) is a scheduled survey to measure the disease burden among the Malaysian population. It used a cross-sectional study design, and national representative respondents were selected via a multistage stratified random cluster sampling. Malaysia was stratified into 13 states and three federal territories. Each state was divided into enumeration blocks. An enumeration block (EB) is a geographical mapping of Malaysia according to the number of living quarters (LQs). Each EB contains around 80 to 120 LQs. The average population is approximately 500 to 600 people. The first stage of sampling was conducted by selecting the enumeration block, and the second stage was the selection of living quarters. Details on this survey’s methodology were explained in the official technical report of NHMS 2019 [4]. Complex sampling design analysis found the prevalence of unawareness of the Malaysian Healthy Plate concept was 83.2%, or equal to 3331 rural adults in Malaysia.

### 2.2. Data Collection

Prior to the study, researchers developed and pre-tested the Malaysia Healthy Plate concept questionnaire to collect information relevant to Malaysian Healthy Plate concept. To verify that each item in the questionnaire is a valid measure of the domain being assessed, the face validity procedure was carried out in three stages, involving experts, researchers, stakeholders, and the technical team. There were two types of the questionnaire: face-to-face interview and self-administered. There were flash cards provided in the form of a code book to assist in the interview.

For the face-to-face interview, the pre-tested questionnaire was bi-lingual (Malay language and English), accompanied by a questionnaire manual prepared as a guide to the data collectors. In order to ensure the message reached various ethnic groups of the community, the self-administered questionnaires were in four languages: Malay, English, Mandarin, and Tamil. Prior to the actual survey, the instrument underwent forward–backward translation of the four languages, and cognitive interview testing among the community, in order to be appropriate for use in the NHMS 2019 among the Malaysian population.

There were flashcards provided in the form of a code book to assist in the interview. The face-to-face interview questionnaire was programmed into an application, and the data collection was performed using tablets. Respondents were given the tablet to fill themselves with the self-administered questionnaires. Hardcopies of the self-administered questionnaires were also prepared should the respondent choose to answer in the paper. Other data also being collected for this study were: (1) sociodemographic characteristics—strata, gender, age, ethnicity, marital status, education level, occupation, income; (2) dietary practice—plain water, fruit, and vegetable intake; and (3) health literacy assessment. The information collected was entered into the IKU Survey Creating System (IKU SCS) using electronic devices, and then submitted to the server in the institute headquarters.

### 2.3. Ethical Approval and Consent to Participate

This study was conducted according to the guidelines laid down in the Declaration of Helsinki, and all procedures involving research study respondents were approved by the Medical Research and Ethics Committee (MREC), Ministry of Health Malaysia, and was registered with the National Medical Research Register (NMRR) with the reference NMRR-18-3085-44207. The date of ethical clearance was 20 December 2018. Written informed consent was obtained from all respondents.

### 2.4. Variables Definitions

Those that were aware of the concept were required to report the Malaysian Healthy Plate concept and provide information on whether or not they were implementing it. In order to assess the visibility of the Malaysian Healthy Plate concept, respondents were asked if they had ever heard of the “Malaysian Healthy Plate,” with binary responses “Yes” or “No”. Those that were aware of the concept were asked to indicate and inform on the Malaysian Healthy Plate concept, as well as whether or not they practiced it. Perception or knowledge of something is closely related to awareness. The ability to accurately describe something experienced or known is commonly regarded as a behavioral indicator of conscious awareness. The dependent variable in this study is unaware, which means respondents that have never heard of or are totally unaware of the Malaysian Health Plate Concept.

### 2.5. Data Analysis

Data were analyzed using the complex samples analysis in the IBM Statistical Package for the Social Sciences (SPSS) software for Windows, Version 26.0. Descriptive analyses were used to measure the prevalence of socio-demographic characteristics and related factors influencing the Malaysian Healthy Plate concept awareness. In order to evaluate the relationship between the variables of interest, a multivariate logistic regression analysis was carried out, and meaningful findings were presented as a modified odds ratio with 95% confidence intervals, significance level at the *p*-value of 0.05, and control for all potential confounding factors.

### 2.6. Availability of Data and Materials

The data were not available to the public, and approvals were obtained/required for access to the data. The data set generated and analyzed during the current study will not be shared. This is to protect and maintain respondents’ anonymity and confidentiality. Even though the data are with the corresponding author, due to the sensitive nature of the information of the respondents provided, the data are kept saved in order not to expose the feelings of the respondents to the public.

## 3. Results

The analysis for this study comprised 3331 rural respondents, representing four fifths of Malaysian rural people (83.2%) who were unaware of the Malaysian Healthy Plate concept. In order to assess the visibility of the Malaysian Healthy Plate concept, respondents were asked if they had ever heard of the “Malaysian Healthy Plate”, with binary responses ‘Yes’ or ‘No’. Those who answered ‘Yes’ will be considered aware (code as 0), while those who answered ‘No’ will be considered unaware (code as 1). The ability to accurately describe something experienced or known is commonly regarded as a behavioral indicator of conscious awareness.

In this study, the dependent variable is unaware, which indicates respondents who have never heard of or are completely unaware of the Malaysian Health Plate concept. Table 1 shows the comparison through the Rao-Scott-adjusted chi-square statistic, which found unawareness was significantly higher in male (91.3%, 95% CI: 89.0–93.1), older age group (91.8%, 95% CI: 89.1–93.), other ethnicities (88.5%, 95% CI: 84.5–91.6), widow/divorced (88.1%, 95% CI: 84.0–91.3), no formal education (95.2%, 95% CI: 90.9–97.6), retires/not working (88.4%, 95% CI: 84.1–91.7), and respondents with a bottom 40% threshold household income (85.0%, 95% CI: 82.6–87.1).

Table 2 shows the unadjusted and adjusted logistic regression analyses of the factors influencing the prevalence of unawareness on the Malaysian Healthy Plate concept among rural adults. Male, age group, other ethnicity, education level, and occupation were factors influencing the prevalence of unawareness.

The odds of being unaware of the Malaysian Healthy Plate concept were 4.12 times higher for males as compared to females. The odds of unawareness were 1.46 times higher among the age of 40 to 59 compared to the age of 18 to 39. The odds of unawareness were 3.47 times higher among no formal education and 2.44 among primary education than tertiary education. The employees in the private sector (2.68), self-employed (2.70), retirees (2.31), and unpaid workers (2.62) were more unaware than government/semi-government employees.

Factors such as ethnicity, marital status, household income threshold, health promotion literacy status, disease prevention literacy status, health literacy status, and health care literacy status did not reveal any significant association with the prevalence of unawareness, despite their significant relationship univariate analysis. Similarly, fruits, vegetables, and plain water intake in this study did not significantly associate with the factors influencing unawareness of the Malaysian Healthy Plate concept.

## 4. Discussion

The “Malaysian Healthy Plate” was implemented in 2016 after the “Malaysia Food Pyramid” was introduced in the Malaysian Dietary Guideline in 2010. The current and simpler concept is a realistic guide to benefit the public to make healthier food choices with the tagline “*Suku-Suku-Separu*” or “Quarter-Quarter-Half” [11]. However, a NHMS 2019 report indicated that only 20.4% (95% CI: 18.93, 22.03), or 1 in 5 respondents, were aware of the Malaysian Healthy Plate concept [4]. The survey also revealed that urban respondents (21.5% (95% CI: 19.64, 23.43)) were more aware of the Malaysian Healthy Plate concept than rural respondents (16.8% (95% CI: 14.74, 18.98)). This reveals that, despite being launched for three years, four in five Malaysians were unaware of the Malaysian Healthy Plate concept, and 83.2% of the population living in rural areas have a low level of awareness of the Malaysian Healthy Plate. This finding is similar to a Survey of Nutritional Status of Filipino Children, and Other Population Groups 2015, which reported unawareness of Filipinos towards the “*Pinggang Pinoy*” healthy food plate as 89.4% one year after introduction to the public. “*Pinggang Pinoy”* is a Filipino Plate Method food guide that uses a familiar food plate model to convey the right food group proportions on a per-meal basis, to meet the body’s energy and nutrient needs of Filipino adults, as well as a simple tool for nutrition advice that only takes about 15 min to teach. This survey also reported that the messages or campaign had not reached the population residing in remote areas of the Philippines [15]. The factors associated with the level of awareness among the rural population was discussed elaborately. The potential explanations for these results will be addressed in the multiple logistic regression model according to the relationship’s magnitude.

This study shows that most elderly males were not aware of the “Quarter-Quarter-Half” campaign compared to females. One study on gender differences in the behavior of health information among the Finnish population also found that females were more involved in, and reported much more, active health-related information searches, paid more attention to possible global pandemics, and were much more attentive to how the items they buy in their daily lives impact their health than males [16]. It is also probable that older people living in rural areas reside mostly alone without special care [17]. Living arrangements are an important component of older persons’ overall quality of life, as they experience life changes such as retirement, death of a spouse, and a decline in health. An older person living alone is defined in Malaysia as an older person (60 years and above) staying permanently alone, in the sense that they are the sole occupants of their dwelling, and generally sleep alone in that dwelling. Living alone is one of the salient factors affecting older persons’ quality of life [18]. Most likely, this factor of living alone also causes them not to go for health treatment at the health clinic, limiting their access to the information provided. Levesque et al. suggested that there might be a difficulty for the population in rural areas to access healthcare facilities due to logistic factors and distance from their home [19]. However, the challenge now is to reinforce the healthcare services to cope with the challenge of shifting demographics, particularly in the elderly population, and the rise of NCD. A more drastic approach needs to be taken to reach this target group. However, research by Papazafiropoulos et al. [20] and Grigorescu et al. [21] revealed a negative relationship between the absence of diabetes and the age of those who live alone.

This study also shows that those with low education levels have a positive relationship with a low level of awareness among rural populations. The findings of Adam et al. showed that the level of education in rural areas is extremely poor [22]. This includes the comparatively low degree of information learning due to the lack of a culture of knowledge passion. This situation also makes them unexposed to the Malaysian Healthy Plate campaign, which has been introduced since 2017. Education for “Quarter-Quarter-Half” was also extended to schools. In schools, this campaign was launched to increase school children’s awareness, and sensitivity to the global community about the importance of healthy eating habits and the practice of a healthy lifestyle [23]. However, it is very likely for older people with no formal education background that this information does not reach them. The level of education also plays a key role in instilling an awareness of health. The beneficial effects of education are pervasive, cumulative, and self-amplifying, increasing throughout life [24].

Malaysian Healthy Plate campaigns have been introduced in different settings, such as health clinics, schools, grocery markets, mass media, health-related campaigns, and an online application called MyNutri. However, because of the unpleasant nature of their jobs, the majority of them disregard the advice given to them, and do not gather information on healthy eating. It probably did not reach the target yet because this study’s results show that awareness of Malaysian Healthy Plate negatively relates to those who work, especially those living in rural areas. Residents in rural areas place more emphasis on earning a living to survive [22]. Financial constraints and livelihood factors for families limit their access to treatment at health clinics, and, at the same time, cause them to be unexposed to information provided by health workers. These working populations are also less involved in community activities, and may even cause them to be unexposed to the Malaysian Healthy Plate campaign. These findings need to be given serious attention, and it is recommended that a thorough investigation be conducted.

In this study, healthy dietary behaviors were evaluated, including the intake of plain water, fruits, and vegetables. The results showed no significant correlation with the level of awareness of the Malaysian Healthy Plate. However, based on the Malaysian Adult Nutrition Survey (MANS) 2014, it was found that a large number (94.0%) of Malaysian adults do not consume enough fruits and/or vegetables (less than five servings a day) as recommended by the Malaysian Dietary Guideline 2010 [25]. The NHMS study’s findings (2011) [26] also showed that only 7.5% of Malaysians consumed an adequate amount of fruits and vegetables. The following NHMS study (2015) [3] showed similar results: 94.0% of Malaysian adults did not consume an adequate amount of fruits and/or vegetables daily. The situation is even more worrying when, in 2019, there was a slight increase, where about 94.9% of adults were found not to consume an adequate amount of fruits and/or vegetables daily [4]. The World Health Organization (WHO) recommends an intake of fruits and/or vegetables of at least five servings per day [27]. Inadequate intake of fruits and vegetables were observed among respondents with lower education achievement and lower household income. This problem may be due to the high cost of fruits and vegetables, as well as the wide availability of unhealthy foods, such as energy-dense foods at affordable prices in the stores [11]. According to the findings of the Lung et al. [28], socio-demographic characteristics, such as household size, income, gender, marital status, age, and education, are significantly related to fruit and/or vegetable intake. The Malaysian Dietary Guidelines (MDG) 2010 [9] recommends that adults consume six to eight glasses of plain water daily. Integrating dietary practices into everyday life may have a profound effect on human health. Poor food consumption choices can increase the risk of certain diseases and conditions, such as diabetes, high blood pressure, and obesity.

In terms of comprehensive health care, rural communities need to be exposed to health care awareness and treatment vital for disease. Although health literacy was not shown to be significant for multivariate analysis in this study, entities are encouraged to acquire and understand basic health information and services and enable them to make the right decisions to maintain and promote their health. Low health literacy is a concern with a high prevalence in the United Kingdom and globally [29]. It has a social gradient in the United Kingdom, with a higher incidence in lower social classes. It is indeed related to a higher prevalence of long-term health problems, a lower self-rated state of health, and more challenges self-managing long-term health problems [29]. NHMS 2019 also revealed 35.1% of adults have low health literacy. Most rural areas are found to have a lack of information on health screening, including undergoing blood pressure tests, and getting healthy eating counselling at health clinics. This vulnerable group also lacks an understanding of the media’s information (internet, newspapers, and magazines) to be healthier [4]. It is also possible that what happened in this study is that the rural population was not good at interpreting the healthy dishes introduced, leading to a lack of awareness of practicing a healthy diet [4]. Improved medical services and practitioner awareness of a patient’s health literacy can help to address these issues.

Future studies should consider undertaking an assessment of such factors to provide a more comprehensive understanding of the broader factors that may impact adopting the “Quarter-Quarter-Half” campaign in the rural setting. As a suggestion for improvement to the existing program, the promotion of healthy eating health is expected to be extended to the grassroots level by approaching the village committee (village head), residents’ associations, and target groups mentioned above. A clearer explanation should be given to patients who come for advice or medication at health clinics on a regular basis. The campaign can also be actively disseminated through mainstream channels such as state television and radio. It is also proposed that future research should employ an approach that investigates whether being aware of the “Healthy Plate” leads to long-term changes in dietary behavior.

### Strengths and Limitations

The study’s strength is that the level of awareness of the Malaysian Healthy Plate concept can be generalized to the entire Malaysian population. These data provide a unique opportunity to evaluate public awareness of the health-related campaigns, and the application of the Malaysian Healthy Plate concept into eating behaviors. This assessment has limitations. This survey assesses only the adult population of the site, and neither the children’s game nor the sections for nutrition professionals. Malaysian Healthy Plate is a work in progress; some of the assessment concerns may be addressed by the time of this paper’s publication. Although these assessments were intended to explore aspects of Malaysian Healthy Plate’s relevance for the population, they cannot substitute for the most critical strategy: involving lower-literate and culturally-diverse users in the design, testing, and iterative revision of the site. One key component might be the related co-morbidities in the examined population, which was also recognized as a study limitation. The survey used a cross-sectional design, and limited the causal relationship between the predictors and the outcome.

## 5. Conclusions

This study found that males, adults aged 40 to 59 years old, those without formal education, and adults with all occupations (except for students) in rural areas had higher odds of unawareness of the Malaysian Healthy Plate concept. Effective strategies to promote the Malaysian Healthy Plate concept are needed in order to reduce the risk of having NCDs among this population. This study suggests that more promotion of the Malaysian Healthy Plate concept is needed in the rural community through local associations or NGOs.

## Figures and Tables

**Figure 1 ijerph-19-06257-f001:**
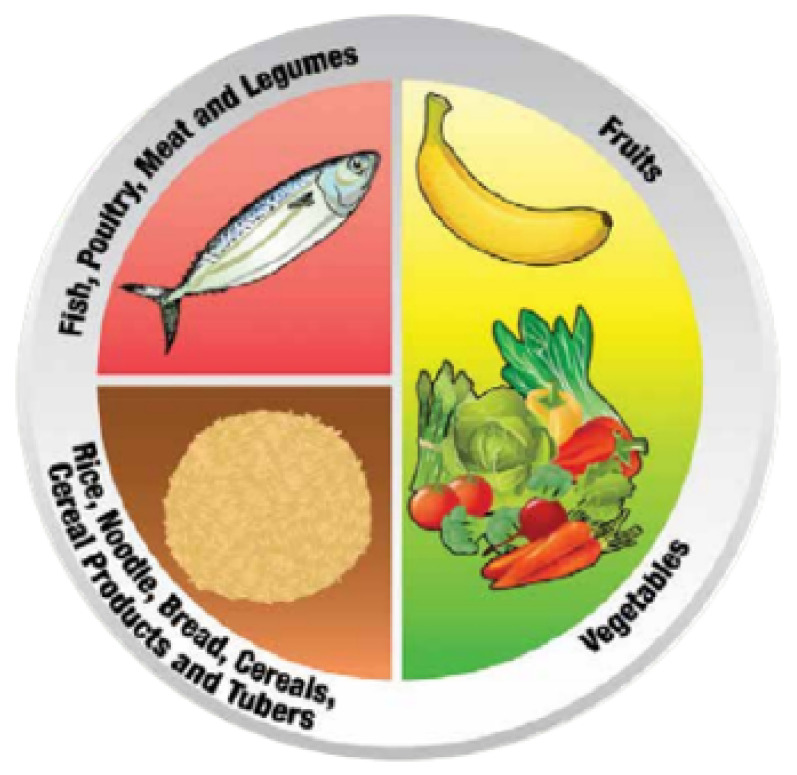
The Malaysian Healthy Plate [11].

**Table 1 ijerph-19-06257-t001:** Prevalence of Malaysian Healthy Plate concept unawareness according to socio-demographic characteristics.

Socio-Demographic Characteristics	Unawareness				
	Unweighted Count	Estimated Population	Prevalence (%)	95% Confidence Interval	
				Lower	Upper
Gender					
Male	1653	2,206,226	91.3	89.0	93.1
Female	1678	1,677,388	74.6	71.5	77.5
Age group					
18–39 years old	1145	2,096,288	80.5	77.2	83.3
40–59 years old	1153	1,061,047	83.6	80.7	86.2
60+ years old	1033	726,278	91.8	89.1	93.9
Ethnicity					
Malays	2441	2,311,548	80.5	77.8	82.9
Chinese	129	79,569	82.4	72.1	89.4
Indians	93	36,675	70.3	61.7	77.7
Others	668	1,455,822	88.5	84.5	91.6
Marital status					
Single	660	1,146,797	83.7	79.9	86.9
Married	2206	2,369,561	82.3	79.8	84.6
Widow/divorced	465	367,256	88.1	84.0	91.3
Education level					
No formal/unclassified	408	503,773	95.2	90.9	97.6
Primary	1036	1,103,626	90.9	88.0	93.1
Secondary	1504	1,770,241	81.0	78.0	83.7
Tertiary	376	498,742	68.3	62.8	73.3
Occupation					
Gov./semi gov.	159	181,930	61.6	52.4	70.0
Private	738	1,133,113	86.0	82.2	89.1
Self-employed	832	979,628	86.8	83.3	89.6
Retirees/not working	837	752,989	88.4	84.1	91.7
Student	57	98,056	80.7	64.2	90.7
Unpaid workers	705	735,921	77.7	73.2	81.6
Threshold household income					
Bottom 40%	2456	2,891,265	85.0	82.6	87.1
Middle 40%	543	650,709	77.7	73.3	81.5
Top 20%	105	102,529	66.9	54.8	77.0

All the results were based on weighted estimates. Unawareness was defined as being unaware or never having heard of the Malaysian Health Plate concept.

**Table 2 ijerph-19-06257-t002:** Factors associated with unawareness of “Malaysian Healthy Plate” in rural areas.

Variables	Complex Sample Logistic Regression Analysis	
	OR (95% Confidence Interval)	aOR (95% Confidence Interval)
Gender		
Male	3.56 (2.73–4.63) *	4.17 (3.09–5.64) *
Female	1	1
Age group		
18–39 years old	1	1
40–59 years old	1.24 (0.99–1.56)	1.46 (1.09–1.96) *
60+ years old	2.74 (1.89–3.96) *	1.63 (0.92.89)
Ethnicity		
Malays	1	1
Chinese	1.13 (0.62–2.06)	0.57 (0.15–2.14)
Indians	0.57 (0.38–0.862) *	0.84 (0.51–1.37)
Others	1.86 (1.28–2.71) *	1.48 (0.97–2.26)
Marital status		
Single	1	1
Married	0.79 (0.66–0.94) *	0.80 (0.58–1.09)
Widow/divorced	1.35 (1.02–1.79) *	0.98 (0.56–1.72)
Education level		
No formal/unclassified	9.28 (4.43–19.42) *	3.84 (1.47–10.04) *
Primary	4.63 (3.23–6.63) *	2.54 (1.52–4.25) *
Secondary	1.99 (1.48–2.66) *	1.30 (0.90–1.89)
Tertiary	1	1
Occupation		
Gov./semi gov.	1	1
Private	3.83 (2.43–6.01 )*	2.68 (1.55–4.65) *
Self-employed	4.09 (2.63–6.37) *	2.70 (1.53–4.77) *
Retirees/not working	4.76 (2.94–7.71) *	2.31 (1.22–4.36) *
Student	2.61 (1.03–6.65) *	2.60 (0.86–7.83)
Unpaid workers	2.18 (1.42–3.33) *	2.62 (1.52–4.50) *
Threshold household income		
Bottom 40%	2.81 (1.68–4.70) *	1.53 (0.81–2.88)
Middle 40%	1.72 (1.04–2.87) *	1.04 (0.55–1.96)
Top 20%	1	1
Fruit intake		
Adequate	1	1
Inadequate	0.89 (0.59–1.33)	0.85 (0.52–1.37)
Vegetable intake		
Adequate	1	1
Inadequate	1.14 (0.73–1.78)	1.31 (0.79–2.17)
Plain water intake		
Adequate	1	1
Inadequate	1.31 (0.98–1.76)	1.29 (0.90–1.84)
Health literacy status		
Limited	2.26 (1.617–3.16) *	1.38 (0.90–2.09)
Sufficient	1.29 (0.95–1.76)	1.22 (0.82–1.81)
Excellent	1	1

OR: odds ratio; aOR: adjusted odds ratio; * significant *p* < 0.05 for complex sample logistic regression.

## Data Availability

The data were not available to the public, and approvals were obtained/required for access to the data.

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
