# Peer review of "Factors Influencing the Levels of Awareness on Malaysian Healthy Plate Concept among Rural Adults in Malaysia"

_ijerph, 2022, doi:10.3390/ijerph19106257_

Round 1

Reviewer 1 Report

This original paper deals with an important issue that is on the WHO agenda of health topics deserving further scientific attention. The study is comprised of an interesting and welcomed analysis of the factors that are associated with the unawareness of a healthy eating model proposed among rural adults in Malaysia.

This research is well done and the methodology and results are well structured and presented.

Regarding the limitations of this study, I think that one important variable could be the associated comorbidities in evaluated people.

Related to “living alone” and its relation with “older persons' quality of life”, I invite the authors to read and consider this paper: https://doi.org/10.3390/ijerph18063249

Author Response

This original paper deals with an important issue that is on the WHO agenda of health topics deserving further scientific attention. The study is comprised of an interesting and welcomed analysis of the factors that are associated with the unawareness of a healthy eating model proposed among rural adults in Malaysia.

This research is well done and the methodology and results are well structured and presented.

Thank you for the comment.

Regarding the limitations of this study, I think that one important variable could be the associated comorbidities in evaluated people.

Added in LIMITATION OF THE STUDY:

One key component might be the related comorbidities in the examined population, which was also recognized as a study limitation.

Related to “living alone” and its relation with “older persons' quality of life”, I invite the authors to read and consider this paper: https://doi.org/10.3390/ijerph18063249

Added to SECTION DISCUSSION: 

However, research by Papazafiropoulos et. al [19] and Grigorescu et. al [20] revealed a negative relationship between the absence of diabetes and age (who additionally found that living alone).

Reviewer 2 Report

The submitted manuscript presents the results of a study aimed to assess factors affecting the Malaysia Healthy Plate campaign's awareness among the rural population. The campaign was launched in 2017 while the data were collected between July and September 2019. In L23-24 the authors state  that (…) About four-fifth (83.2%) of rural adults in Malaysia were unaware of the Malaysian Healthy Plate concept after three years of implementation. This immediately raises the question what kind of promotional activities had been carried out in 2017-2019 to raise the awareness in both rural and urban areas? This should be introduced in the paper’s Introduction and further elaborated on in the Discussion.

The paper is clear and reads well but in some areas lacks depth – for example it would be interesting for Readers to learn if although the concept was not well known in 2019 – have other activities been conducted in 2020 and 2021? Or had for example Covid halted this? Another question that arises when reading the Discussion is why fruit and vegetable consumption is so low in the country?

L105 and L160  state that the sample represents 3,883,614 population in Malaysia. Please explain what is this number. The total population of Malaysia surpassed 32 million in 2019.

Author Response

What kind of promotional activities had been carried out in 2017-2019 to raise the awareness in both rural and urban areas? 

Added to SECTION INTRODUCTION:

Malaysian Healthy Plate campaigns have been introduced in different settings such as health clinics, schools, grocery markets, mass media, health-related campaigns, and an online application called MyNutri (MOH, 2019). It is predicted that this concept will positively change citizens' dietary habits. However, the Malaysian may experience less exposure to this campaign or lack understanding of the Malaysian Healthy Plate concept. Continuous education and promotion through various channels and multiagency collaboration on the implementation are recommended to ensure this healthy eating concept can deliver effectively to the public, especially now during the COVID-19 pandemic.

Have other activities been conducted in 2020 and 2021? Or had for example Covid halted this? 

No other activities been conducted in 2020 and 2021.

Another question that arises when reading the Discussion is why fruit and vegetable consumption is so low in the country?

Added to SECTION DISCUSSION:

According to the findings of the Lung et. al, sociodemographic characteristics such as household size, income, gender, marital status, age, and education are significantly related to fruit and/or vegetable intake.

L105 and L160  state that the sample represents 3,883,614 population in Malaysia. Please explain what is this number. The total population of Malaysia surpassed 32 million in 2019.

Thank you, I amended the sentence “A total of 3,331 rural population were included in the analysis, representing 3,883,614 population in Malaysia” into "A total of 3,331 rural population were included in the analysis which represents about 21.3 million adults 18 years old in Malaysia"

Reviewer 3 Report

The aim of the study was to assess the factors that influence the awareness of the Malaysian Healthy Plate among the rural population in Malaysia. The study is properly designed.

The way in which the results are presented and discussed is unquestionable. The discussion of the results is also logically structured and comprehensively presented. The authors also pointed out the strengths and weaknesses of the studies carried out. There are no concerns about form and content. Congratulations!

The only thing missing is the reference in Figure 1.

Author Response

The aim of the study was to assess the factors that influence the awareness of the Malaysian Healthy Plate among the rural population in Malaysia. The study is properly designed.

The way in which the results are presented and discussed is unquestionable. The discussion of the results is also logically structured and comprehensively presented. The authors also pointed out the strengths and weaknesses of the studies carried out. There are no concerns about form and content. Congratulations!

Thank you for the comment.

The only thing missing is the reference in Figure 1.

Added reference for FIGURE 1: 

Ministry of Health. Malaysian Healthy Plate Guideline. National Coordinating Committee on Food and Nutrition; Ministry of Health Malaysia: Kuala Lumpur, Malaysia, 2016.

Reviewer 4 Report

1 – Line 22 to 23 – “A total of 3,331 rural population were included in the analysis, representing 3,883,614 population in Malaysia. Review the data and the form of presentation.

2 – Line 66 – “meat and vegetables' what are vegetables? Could it be legumes (beans)?

3 – Line 96 to 99 – What is the justification for individuals “the selected living quarters”. I think the term living is not the most appropriate.

4 – Line 105 - A total of 3,331 rural population were included in the analysis, representing 3,883,614 population in Malaysia. Review the data and the form of presentation.

5 – Line 108 to 109 - Sample selection starts at Enumeration Block (EB) to the Living Quarters (LQ) and finally to the individual residing in the living quarters. Make it clearer.

6 – Line 113 to 121 – What is the justification for adopting several languages ​​and different languages ​​for the questionnaires (“For the face-to-face interview, the pre-tested questionnaire was bi-lingual (Malay language and English) and The self -administered questionnaires were 117 in four languages; Malay, English, Mandarin, and Tamil”)? Were reverse translations performed for all instruments? Have all instruments been validated?

7 - Line 136 to 142 - Confusing text. Review English language.

8 – Add the description of the technique used for content validation and criterion validation. Was a judge technique performed? Was the Delphi technique used? If yes, what % was used as a cut-off point?

9 – Line 152 to 157 – The authors claim that the data was not publicly available because it was sensitive. Was the data treated individually, so that it was possible to identify the respondents? What data was considered sensitive?

10 – Line 160 to 170 – It is not possible to understand how 3,331 can represent 3,88,614. All numerals need correction. Some numerals have a comma and some have a dot.

11 - In Table 1 - Is the variable Unawere or unawareness Briefly describe the concept adopted. Psychometric instruments require language that adopts concepts from Psychology.

12 – In Table 1 - The prevalence equation is: Prevalence = (No. of cases No)/ (of individuals in the study) x constant value. Checking what is in the table, I understand that the authors refer to the % of affirmations. Collect and correct.

13 – Table 2 also needs review review concepts to confirm the data.

14 – I think the authors should add a summary description of the content of 'Pinggang Pinoy', the Filipino healthy food plate.

Author Response

1 – Line 22 to 23 – “A total of 3,331 rural population were included in the analysis, representing 3,883,614 population in Malaysia. Review the data and the form of presentation.

Amended:

A total of 3,331 rural population were included in the analysis which represents about 21,300,000 adults 18 years old in Malaysia.

5 – Line 108 to 109 - Sample selection starts at En

2 – Line 66 – “meat and vegetables' what are vegetables? Could it be legumes (beans)?

The Malaysian Healthy Plate consumption of a quarter plate of grains, grains products, cereals products and tubers; a quarter plate of fish, poultry, meat and legumes; half a plate of fruits and vegetables.

3 – Line 96 to 99 – What is the justification for individuals “the selected living quarters”. I think the term living is not the most appropriate.

The NHMS 2019 covered both urban and rural areas in all 13 states and 3 federal territories in Malaysia. The target population was the residence in the non-institutional living quarters (LQs). Institutional population such as those staying in hotel, hostels, hospitals, etc. were excluded from this survey.

4 – Line 105 - A total of 3,331 rural population were included in the analysis, representing 3,883,614 population in Malaysia. Review the data and the form of presentation.

Amended:

A total of 3,331 rural population were included in the analysis which represents about 21,300,000 adults 18 years old in Malaysia.

5 – Line 108 to 109 - Sample selection starts at Enumeration Block (EB) to the Living Quarters (LQ) and finally to the individual residing in the living quarters. Make it clearer.

Amended:

The National Health and Morbidity Survey (NHMS) is a scheduled survey to measure the disease’s burden among Malaysian population. It used a cross-sectional study design and national representative respondents were selected via a multistage stratified random cluster sampling. Malaysia was stratified into 13 states and three federal territories. Each state was divided into enumeration blocks. Enumeration block (EB) is a geographical mapping of Malaysia according to the number of living quarters (LQs). Each EB and it contains around 80 to 120 LQs. The average population is approximately 500 to 600 people. The first stage of sampling was con-ducted by selecting the enumeration block, and the second stage was the selection of living quarters. Details on this survey's methodology were explained in the official technical report of NHMS 2019.

6 – Line 113 to 121 – What is the justification for adopting several languages ​​and different languages ​​for the questionnaires (“For the face-to-face interview, the pre-tested questionnaire was bi-lingual (Malay language and English) and The self -administered questionnaires were 117 in four languages; Malay, English, Mandarin, and Tamil”)? Were reverse translations performed for all instruments? Have all instruments been validated?

Added in SECTION MATERIAL & METHOD:

In order to ensure the message reached various ethnic groups of the community, the self-administered questionnaires were in four languages; Malay, English, Mandarin, and Tamil. Prior to the actual survey, the instrument underwent forward-backward translation of the four languages and cognitive interview testing among the community in order to be appropriate for use in the NHMS 2019 among the Malaysian population.

7 - Line 136 to 142 - Confusing text. Review English language.

Amended:

Those that were aware of the concept were required to report a healthy plate concept and provide information on whether or not they were implementing it. In order to assess the visibility of the Malaysian Healthy Plate concept, respondents were asked if they had ever heard of the "Malaysian Healthy Plate," with binary responses 'Yes' or 'No.' Those that were aware of the concept were asked to indicate and inform on the Malaysian Healthy Plate concept, as well as whether or not they practiced it.

8 – Add the description of the technique used for content validation and criterion validation. Was a judge technique performed? Was the Delphi technique used? If yes, what % was used as a cut-off point?

Added to SECTION MATERIAL & METHOD:

To verify that each item in the questionnaire is a valid measure of the domain being assessed, the face validity procedure was carried out in three stages involving experts, researchers, stakeholders, and the technical team. In terms of instrument reliability, this questionnaire's key domains all have Cronbach's alpha values better than 0.7.

9. Line 152 to 157 – The authors claim that the data was not publicly available because it was sensitive. Was the data treated individually, so that it was possible to identify the respondents? What data was considered sensitive?

The National Institute of Health, Ministry of Health Malaysia (data ethics committee) has placed restriction on sharing the full dataset due to cases involving researchers manipulating the data. The authors also confirm they did not have any special access privileges that others would not have for the NHMS
2019 data. Sensitive nature of the information of the respondents provided such as biometric data for the purpose of uniquely identifying a respondent and also data concerning health. This data are kept saved in order not to expose the feelings of the respondents to the public.

10 – Line 160 to 170 – It is not possible to understand how 3,331 can represent 3,88,614. All numerals need correction. Some numerals have a comma and some have a dot.

Amended:

A total of 3,331 rural population were included in the analysis which represents about 21,300,000 adults 18 years old in Malaysia.

11 - In Table 1 - Is the variable Unawere or unawareness Briefly describe the concept adopted. Psychometric instruments require language that adopts concepts from Psychology.

12 – In Table 1 - The prevalence equation is: Prevalence = (No. of cases No)/ (of individuals in the study) x constant value. Checking what is in the table, I understand that the authors refer to the % of affirmations. Collect and correct.

13.  Table 2 also needs review review concepts to confirm the data.

For no 11, 12, & 13:

Data were analyzed using the complex samples module using the IBM Statistical Package for the Social Sciences (SPSS) software for Windows, Version 26.0. The prevalence of unawareness, interpretation, and practicing status were calculated using complex sample frequencies. Rao-Scott adjusted chi-square statistic was used for the test if independent between the unawareness status and sociodemographic characteristics. The associations between unawareness to other variables were evaluated using complex sample logistic regression, with 95% confidence intervals, significance level at the p-value of 0.05, and control for all possible confounding factors.

14.  I think the authors should add a summary description of the content of 'Pinggang Pinoy', the Filipino healthy food plate.

Added to SECTION DISCUSSION:

Pinggang Pinoy is a Filipino Plate Method food guide that uses a familiar food plate model to convey the right food group proportions on a per-meal basis, to meet the body's energy and nutrient needs of Filipino adults, as well as a simple tool for nutrition advice that only takes about 15 minutes to teach. 

Round 2

Reviewer 4 Report

1 – Line 22 to 23 – “A total of 3,331 rural population were included in the analysis, representing 3883,614 population in Malaysia. Review the data and the form of presentation.

  • A total of 3,331 rural population were included in the analysis which represents about 21,300,000 adults 18 years old in Malaysia.

R- The error remains the same. Numerals must be separated by dots. In addition, the authors must state what % of the rural population the study sample represents and what the statistical criterion for the sample size is.

2 – Line 66 – “meat and vegetables' what are vegetables? Could it be legumes (beans)?

2.1 The Malaysian Healthy Plate of a quarter plate of grains, grains products, cereals products and tuber; a quarter plate of fish, poultry, mean and legumes, Half a plate of fruits ande vegetables.

R – The authors did not answer the question: what are vegetables? Could it be vegetables (beans)?

8 – Add the description of the technique used for content validation and criterion validation. Was a judge technique performed? Was the Delphi technique used? If yes, what % was used as a cut-off point?

8.1 To verify that each item in the questionnaire is a valid measure of the domain being assessed, the face validity procedure was carried out in three stages involving experts, researchers, stakeholders, and the technical team. In terms of instrument reliability, this questionnaire's key domains all have Cronbach's alpha values better than 0.7.

R – Add the exact value of Cronbach's Alpha and if there is more than one factor for the construct, inform the values.

11 - In Table 1 - Is the variable Unawere or unawareness Briefly describe the concept adopted. Psychometric instruments require language that adopts concepts from Psychology.

A - It was not answered.
